# Improvement of Symptoms and Cardiac Magnetic Resonance Abnormalities in Patients with Post-Acute Sequelae of SARS-CoV-2 Cardiovascular Syndrome (PASC-CVS) after Guideline-Oriented Therapy

**DOI:** 10.3390/biomedicines11123312

**Published:** 2023-12-14

**Authors:** Mariann Gyöngyösi, Ena Hasimbegovic, Emilie Han, Katrin Zlabinger, Andreas Spannbauer, Martin Riesenhuber, Kevin Hamzaraj, Jutta Bergler-Klein, Christian Hengstenberg, Andreas Kammerlander, Stefan Kastl, Christian Loewe, Dietrich Beitzke

**Affiliations:** 1Division of Cardiology, 2nd Department of Internal Medicine, Medical University of Vienna, 1090 Vienna, Austria; ena.hasimbegovic@meduinwien.ac.at (E.H.); emilie.han@meduniwien.ac.at (E.H.); katrin.zlabinger@meduniwien.ac.at (K.Z.); andreas.spannbauer@meduniwien.ac.at (A.S.); martin.riesenhuber@meduniwien.ac.at (M.R.); kevin.hamzaraj@meduniwien.ac.at (K.H.); jutta.bergler-klein@meduniwien.ac.at (J.B.-K.); christian.hengstenberg@meduniwien.ac.at (C.H.); andreas.kammerlander@meduniwien.ac.at (A.K.); stefan.kastl@meduniwien.ac.at (S.K.); 2Department of Biomedical Imaging and Image-Guided Therapy, Medical University of Vienna, 1090 Vienna, Austria; christian.loewe@meduniwien.ac.at (C.L.); dietrich.beitzke@meduniwien.ac.at (D.B.)

**Keywords:** long COVID, COVID-19, PASC-CVS, cardiac magnetic resonance imaging, CMR, myopericarditis, chronic pericardial effusion

## Abstract

Cardiac magnetic resonance (CMR) studies reported CMR abnormalities in patients with mild–moderate SARS-CoV-2 infection, suggesting ongoing myocardial inflammation. Patients (*n* = 278, 43 ± 13 years, 70.5% female) with post-acute sequelae of SARS-CoV-2 cardiovascular syndrome (PASC-CVS) were included prospectively into the Vienna POSTCOV Registry between March 2021 and March 2023 (clinicaltrials.gov NCT05398952). Clinical, laboratory, and CMR findings were recorded. Patients with abnormal CMR results were classified into isolated chronic pericardial (with/without pleural) effusion, isolated cardiac function impairment, or both (myopericarditis) groups. Medical treatment included a nonsteroidal anti-inflammatory agent (NSAID) for pericardial effusion and a condition-adapted maximal dose of heart failure (HF) treatment. Three months after medical therapy, clinical assessment and CMR were repeated in 82 patients. Laboratory analyses revealed normal hematological, inflammatory, coagulation, and cardiac biomarkers. CMR abnormalities were found in 155 patients (55.8%). Condition-adapted HF treatment led to a significant increase in the left ventricular ejection fraction (LVEF) in patients with initially reduced LVEF (from 49 ± 5% to 56 ± 4%, *p* = 0.009, *n* = 25). Low–moderate doses of NSAIDs for 3 months significantly reduced pericardial effusion (from 4/3;5.75/mm to 2/0;3/mm, median/interquartile ranges/*p* < 0.001, *n* = 51). Clinical symptoms improved markedly with a decrease in CMR abnormalities, which might be attributed to the maintenance of NSAID and HF medical treatment for PASC-CVS.

## 1. Introduction

Cardiovascular symptoms, such as arrhythmias, exercise-induced dyspnea, chest pain, and cardiac fatigue syndrome, are common in patients with long COVID syndrome (post-acute sequelae of SARS-CoV-2 infection cardiovascular symptoms, PASC-CVS) [1,2,3]. Several studies have reported cardiac abnormalities detected by cardiac magnetic resonance imaging (CMR) in many non-hospitalized SARS-CoV-2-infected patients several months after the acute infection [4,5,6,7]. However, such findings (e.g., nonsignificant pericardial effusion or mildly enlarged left and/or right ventricle) were generally considered clinically insignificant, not requiring further treatment [4,5,6,7]. Even if idiopathic chronic mild pericardial effusion has a good prognosis with rare complications [8], the risk of developing chronic pericarditis or a deterioration in heart function with consequences after SARS-CoV-2 infection is currently not calculable. Furthermore, 6 month mortality was significantly higher in SARS-CoV-2-infected patients if they had pericarditis compared with COVID-19-positive patients without pericarditis, even with a small amount of pericardial effusion [9,10]. Additionally, recent findings underlie the presence of ongoing myocardial tissue inflammation due to dysregulated immune system components [3,4], indicating low-dose anti-inflammatory maintenance therapy with steroids combined with angiotensin-receptor blocker (ARB) (MYOFLAME study, NCT05619653). In the absence of active inflammation with normal laboratory values for the inflammatory parameter, we treated symptomatic patients with PASC-CVS displaying abnormal CMR findings in accordance with the current guidelines for cardiac dysfunction and with the maintenance of low-dose nonsteroidal anti-inflammatory drugs (NSAID) in case of chronic pericardial effusion.

The aim of our study was to investigate the effects of our treatment regimen on cardiovascular symptoms and abnormalities found by CMR in symptomatic patients with PASC-CVS.

## 2. Materials and Methods

### 2.1. Study Design

Our POSTCOV study is an ongoing prospective registry study (ClinicalTrials.gov Identifier: NCT05398952). The presented methods and results conform with the STROBE guidelines [11].

Patients with PASC-CVS and CMR scans were included prospectively in the Vienna POSTCOV Registry between March 2021 and March 2023. Written informed consent was obtained from all patients before study entry. The study was approved by the local Ethical Committee of the Medical University of Vienna, Austria (EC: 1008/2021 and 1758/2022), and was performed in accordance with the Declaration of Helsinki. Clinical data, CMR findings, and blood sampling results were recorded. CMR was performed if clinically indicated by chest pain, persisting cough, ongoing subfebrility, palpitation, orthostatic intolerance, or ECG abnormalities with suspected chronic myopericarditis. CMR was repeated after medical therapy if clinically justified (Figure 1).

### 2.2. Inclusion and Exclusion Criteria

Patients with PASC-CVS [12,13,14] were included if they had previous mild or moderate COVID-19 infection confirmed by quantitative real-time polymerase chain reaction (PCR) and were not hospitalized during the acute illness, had no actual or previous systemic diseases (e.g., systemic inflammatory, rheumatic, oncological, cardiovascular, or renal illnesses), and had at least three symptoms from three different organs [1,3]. The main exclusion criteria were signs of active infection with elevated inflammatory parameters (e.g., C-reactive protein, leukocytes, or fibrinogen), missing COVID-19 PCR test, and systemic disease, as well as reasons for secondary pericardial effusion (e.g., traumatic, drug-induced, pulmonary hypertension, metabolic, amyloidosis, rheumatic, or oncologic).

### 2.3. Clinical Data

Clinical data were collected during the outpatient visit and included detailed anamnesis, including age, sex, cardiovascular risk factors (hypertension, diabetes, hyperlipidemia, or smoking), previous or current systemic disease, current medical treatment, time of COVID positivity, time between SARS-CoV-2 infection and CMR (days), time between first COVID-19 vaccine and CMR (days), and number of patients with past SARS-CoV2 infection with probable lasting immunity and at least one COVID-19 vaccine.

### 2.4. Laboratory Data

Blood sampling was performed at the first outpatient visit and at the follow-up. The clinical laboratory data included hematological, inflammatory, coagulation, and cardiac markers. In addition, QuantiFERON-TB Gold and Borrelia tests were performed if clinically indicated. The laboratory examinations were performed at the Department of Laboratory Medicine, Medical University of Vienna, Wien, Austria [15]. The laboratory methods can be found at the institution’s homepage (https://www.akhwien.at/default.aspx?pid=3985, accessed on 1 March 2021).

### 2.5. CMR Acquisition and Analysis

All CMR examinations were performed using either 1.5T or 3T MR systems (Philips Ingenia, Eindhoven, The Netherlands, and Siemen Avanto Fit, Siemens Vida, Munich, Germany), with dedicated protocols to screen for inflammation according to the SCMR guidelines [16,17]. All CMR protocols included short-axis cine images for the evaluation of cardiac function, edema-sensitive sequences for the detection of myocardial edema, and late gadolinium enhancement (0.15 mmol/kg gadobutrol/Gadovist; Bayer Vital GmbH, Leverkusen, Germany/if the estimated glomerular filtration rate was >30 mL/min/1.73 m^2^) sequences for the detection of myocardial scarring. Image postprocessing and reporting was conducted by experienced cardiac imaging specialists according to recent guidelines [18]. Briefly, we used a stack of short-axis SSFP cine views to determine the end-diastolic volumes (EDVs, mL) and end-systolic volumes (ESVs, mL) of the left (LV) and right ventricle according to standard protocols [16]. The ejection fraction (EF, %) was calculated as the difference between EDV and ESV (stroke volume) divided by EDV and given as a percentage. T1 parametric mapping was performed using a modified Look–Locker inversion recovery (MOLLI) sequence as described previously [19]. The presence and quantification of pericardial and pleural effusion were determined in SSFP cine views, black-blood sequences, and parametric mapping slices as appropriate, and the amount was measured at the largest diameter and given as millimeters.

### 2.6. Definitions

The definition of chronic pericardial effusion was in accordance with the ESC and ACC/AHA guidelines [20,21]. Briefly, chronic hemodynamically nonsignificant pericardial effusion was diagnosed if the patient had mildly or moderately sized circumferential pericardial fluid longer than 1 months after the SARS-CoV-2 infection. Cardiac morphologic and/or functional impairment was defined based on guidelines [22,23] and involved monolateral or bilateral enlargement of the ventricles with/without a decrease in monolateral or bilateral ventricular function or myocardial edema, T1 signal increase, or late gadolinium enhancement. Myopericarditis was diagnosed if the patient had both chronic pericardial effusion and cardiac morphological and/or functional impairment [20].

### 2.7. Treatments

Patients with CMR abnormalities were divided into three main groups: isolated chronic pericardial effusion with/without pleural effusion; signs of post-COVID myocarditis with/without ventricular systolic dysfunction or enlargement of the ventricles or the morphological abnormalities described above; and combined myopericarditis.

Patients with reduced left ventricular systolic function with/without monoventricular or biventricular enlargement or normal systolic function with enlarged ventricles were treated in accordance with the relevant heart failure (HF) guidelines [9,10], including with beta blockers, angiotensin-converting enzyme (ACE) inhibitors, or ARB, with/without diuretics (hydrochlorothiazide, HCT) and aldosterone antagonists (Figure 2). As our patients had no previous cardiac or other comorbidities, and the majority of the patients were middle-aged women, a blood-pressure- and condition-adapted maximal tolerated dose of HF therapy was applied.

Patients with chronic pericardial effusion with/without pleural effusion received NSAIDs at low–moderate doses, in most cases ibuprofen at a 2 × 200 mg daily oral dose for 3 months, with H2-receptor antagonists at a daily dose of 20 or 40 mg if necessary (Figure 2).

This treatment was established based on the guidelines for chronic pericardiac effusion therapy [7,8] (Class I, Level C) and a routine clinical treatment regimen based on the literature [8,24] because there are no evidence-based therapy guidelines for idiopathic (probably post-viral) chronic hemodynamically nonsignificant pericardiac effusion without signs of acute inflammation in symptomatic patients. Our combined guideline-oriented and literature-based therapy was established based on the following facts: (1) because all the inflammatory parameters were in the normal range for all patients, high-dose NSAID therapy, eventually combined with colchicine or corticosteroids, was not indicated; (2) three patients had received colchicine previously for 3 months, prescribed at the primary care level, without any efficacy; (3) short-term moderate or high-dose treatment (2 to 4 weeks) did not result in any changes in pericardial effusion controlled by CMR in some patients; (4) there are no evidence-based data on the optimal treatment duration for chronic, hemodynamically nonsignificant but symptomatic pericardial effusion [8,24]; (5) the ongoing MYOFLAME study suggests anti-inflammatory cardioprotective treatment for 4 months (NCT05619653).

Patients with both pericardial effusion and cardiac dysfunction were treated with a combination of the abovementioned therapies.

### 2.8. Statistics

Continuous variables were tested for normal or nonnormal distribution and expressed as mean ± standard deviation or median with interquartile range (IQRs), respectively. Nominal variables were categorized as frequencies. Baseline and follow-up data were compared using the two-sided Student’s *t*-test with repeated measurements (normally distributed variables) or the nonparametric Wilcoxon test or the chi-square test for nominal variables. Statistical significance was defined as *p* < 0.05.

## 3. Results

We included a total of 278 patients (43 ± 13 years, 70.5% female). The main cardiac symptoms were reported in patients with indications for CMR imaging: chest pain: *n* = 187 (67.3%), dyspnea: *n* = 153 (55.0%), palpitation: *n* = 147 (52.9%), tachycardia: *n* = 92 (33.1%), thoracic discomfort with/without cough: *n* = 177 (63.9%), reduced physical activity with/without post-exertional malaise: *n*: 166 (59.7%), orthostatic incompetence: *n* = 33 (11.9%).

Laboratory analyses did not reveal elevated acute inflammation; hematological, inflammatory, or coagulation parameters; or cardiac biomarkers (Table 1).

CMR abnormalities were found in 155 patients (55.8%) (Table 2). In total, 58 male (37.4%) and 97 female patients 62.6%) exhibited CMR abnormalities (*p* = 0.001). There were no differences between patients with/without pathological CMR findings regarding clinical (age, time to COVID infection, cardiovascular risk factors, blood pressure, heart rate), ECG, or laboratory parameters.

Appendix A shows the detailed CMR data in the patients with/without CMR abnormalities and the data of the CMR phenotype groups.

The time analysis showed the highest incidence of CMR abnormalities 3–5 months after acute SARS-CoV-2 infection, with persistence of cardiac CMR abnormalities even over 24 months post COVID-19 infection (Figure 3).

Among the patients with CMR abnormalities (*n* = 155), 74 of them (47.7%) received NSAIDs (for chronic pericardial effusion), 81 (52.3%) ARB, 25 (16.1%) ARB/HCT, 6 (3.9%) ACE inhibitor, 68 (24.5%) beta blockers, 15 (5.4%) aldosterone antagonists (for cardiac impairment), and 71 (25.5%) H2-receptor blocker therapy.

All patients with abnormal CMR findings were controlled 3–4 months after the initial visit, and 82 patients underwent a follow-up CMR scan 131 ± 52 days after treatment start.

After medical therapy, the clinical symptoms improved markedly (Figure 4), with a decrease in CMR abnormalities (Table 3 and Figure 4 and Figure 5) in terms of a decreasing amount of pericardial effusion (from 4/3;5.75/mm to 2/0;3/mm, median/IQRs/*p* < 0.001, *n* = 51) and a significant increase in the LVEF of patients with ventricular functional impairment with/without pericardial fluid at first clinical presentation (from 49 ± 5% to 56 ± 4%, *p* = 0.009, *n* = 25).

Table 4 summarizes the detailed changes in CMR abnormalities after 3 month therapy in the different CMR phenotype groups.

## 4. Discussion

To the best of our knowledge, this is the first report on the treatment of patients with PASC-CVS based on CMR findings for chronic hemodynamically nonsignificant pericardial effusion with/without pleural effusion (polyserositis), isolated cardiac morphology and/or function impairment, or combined myopericarditis. As no causative therapy for long COVID-19 syndrome exists, we started the therapy by dividing the patients into three main groups and treated the disease entities with a guideline-oriented medical regimen, supplemented by literature-based and clinical routine therapy if the patients had chronic pericardial effusion.

CMR abnormalities were found in 55.8% of patients with cardiac/cardiovascular complaints. This number is in accordance with other reports stating that patients with COVID-19 exhibited abnormal CMR findings at rates of 18% to 78% [3,4,5,6], depending on the included patient cohort, the severity of the acute infection, hospitalization, and time after the viral infection. Our time analysis revealed a decrease in the prevalence of abnormal CMR findings after 3 to 5 months, probably due to repeated vaccination or to the less cardio-invasive SARS-CoV-2 variants (e.g., Omicron); these findings are in line with other reports [5,25]. However, the role of spontaneous improvement in cardiac abnormalities and the role of an arbitrary intake of diverse anti-inflammatory or antioxidant dietary supplements during the long COVID-19 period cannot be excluded. Puntmann et al. reported follow-up CMR findings at a median of 329 days after the first CMR and found no change in the LVEF (from 56.6 ± 4.6 to 56.9 ± 4.8%) and a significant increase in the RVEF (from 54.0 ± 5.6 to 55.4 ± 5.6%) without supplementary information on specific cardiac therapy [5].

The annual incidence of acute pericarditis before the outbreak of COVID-19 was 0.027% [8] and that of the common viral myocarditis was 0.001–0.01% of the general population [26]. Without evoking unnecessary anxiety, the relatively high incidence of chronic pericardial effusion and myocardial injury persisting for a long time in approximately 10% of the world population infected with SARS-CoV-2 virus requires attention [27,28].

At clinical presentation, acute viral myocarditis was excluded in all patients, based on normal cardiac enzymes, normal inflammatory parameters, and the lack of ECG signs or clinical symptoms of acute myocarditis. Few patients presented isolated myocardial edema, T1 increase, or nonischemic late gadolinium enhancement in CMR imaging, indicating chronic myocardial injury. However, through the lack of supportive acute clinical scenarios or laboratory signs, these CMR findings did not fulfill the modified Lake Louise criteria for acute myocarditis [29,30]. In addition, no CMR was performed during the acute phase of the SARS-CoV-2 infection.

Polyserositis is characterized by inflammation and effusion of the serous membranes, e.g., the pleura, pericardium, and peritoneum. Combined pericarditis and pleuritis is the most common appearance of polyserositis, although diagnosis is difficult, with a lack of diagnostic and therapeutic guidelines [31,32]. Some case reports emphasize the diagnostic challenge of polyserositis in patients, especially in children with multiorgan inflammatory syndromes with acute SARS-CoV-2 infection [33]. We have detected simultaneous pericardial and pleural effusion in 13 patients. The long-time maintenance of polyserositis after the acute infection suggests either a chronic inflammation or autoimmune reaction; both processes require clinical controls.

In general, in the case of chronic pericardial effusion without definitive etiology (supposed SARS-CoV2-induced chronic pericardial effusion) and a lack of signs of systemic inflammation, diverse treatment regimens are recommended [20,21,24]. Less debatable is the treatment of morphological or functional cardiac injury. Patients with pericardial effusion received low or moderate doses of NSAIDs, combined with H2-receptor blocker antagonists if necessary. In accordance with the guideline definitions, patients had the diagnostic criteria for chronic pericardial effusion [20] without clinical (typical pericardial friction rubs) or imaging-proven (ECG or CMR) signs of acute pericarditis or acute viral infection. High doses of NSAIDs are recommended if inflammatory markers are elevated [9,10,20]. However, our patients had no elevated inflammatory or cardiac biomarkers at their first clinical presentation and had symptoms for a longer time, starting after the SARS-CoV-2 infection. In accordance with the ESC guidelines, a lower effective dose may eventually be applied for a shorter period [20]. However, longer NSAID therapy was associated with the decreased recurrence of idiopathic pericardial effusion [24]. Notably, as the figure shows, 3 months of a low–moderate dose of NSAID therapy decreased the amount of pericardial effusion, and it disappeared in several, even if not all, patients. Further steps of treatment with glucocorticoids were not justified in patients with no manifest acute inflammatory pericarditis. An explorative pericardial biopsy was not indicated due to its lack of consequences for decision making about the medical therapy [34].

Patients classified to the functional impairment group received the guideline-oriented HF treatment [22,23]. We observed an improvement in single or biventricular enlargement and/or single or biventricular function in almost all cases. Patients exhibiting myopericarditis were treated with both medical regimens. Their cumulative CMR abnormalities decreased significantly, in parallel with an improvement in symptoms, justifying their indication for treatment.

Most of the patients vaccinated against SARS-CoV-2 received their vaccine before the infection. Theoretically, the vaccine may also induce myocarditis, but its incidence is orders of magnitude lower than that of viral myocarditis [26].

Our study has several limitations. First, a routine CMR is not recommended in patients after COVID-19 disease without cardiac symptoms due to the occasional overinterpretation of nonsignificant CMR changes with questionable relation to previous SARS-CoV-2 infection, except in cases of suspected chronic pericardial effusion (Class IIa, Level C recommendation) [7]. In addition, previous data demonstrated the presence of non-physiological pericardial effusion in up to 30% of clinically asymptomatic patients [4,5]. In accordance with the consensus expert panel recommendation, a CMR should only be performed if it contributes to clinical decision making [32,35]. However, all of our patients presented PASC-CVS.

Several viruses, especially the Epstein–Barr virus, can be reactivated during coronavirus infection [36,37,38]. We cannot exclude the role of reactivated concomitant cardiotropic viruses (e.g., herpes virus, cytomegalovirus) causing chronic pericardial effusion and myopericarditis, even without signs of acute viral infections. Virus diagnostics from the pericardial punctatum may be informative, but with high risk and cost for low benefit and presumptively inadequate information. In addition, the decrease in or disappearance of the pericardial fluid after NSAID treatment suggests a rather autoreactive immune process. In contrast with other CMR studies [39], all of our patients were home-quarantined with no medical record and a lack of information on inflammatory parameters, cardiac enzymes (e.g., troponin T or I), ECG, and echocardiography during the active phase of SARS-CoV-2 infection. Therefore, we cannot correlate real SARS-CoV-2-induced myopericarditis with the CMR findings. However, our patients reported typical PASC-CVS, which was not experienced before the COVID-19 disease. Eighty-two patients agreed with control CMR images after symptom- and CMR-abnormality-oriented treatments. The other patients, with pathological CMR findings, also underwent the medical therapy described above, which led to subjective wellbeing, making the control CMR clinically unnecessary.

Due to the lack of a control or placebo group, the efficacy of the suggested therapy might be overestimated. However, many patients had persistent CMR abnormalities for even longer than 1 year, which were resolved or improved after therapy, in parallel with the decrease in cardiovascular symptoms in our cohort, which suggests the beneficial effect of our therapy. Considering the psychological vulnerability of the patients with PASC-CVS, a blinded study with eventual randomization to a placebo arm was not accepted by our patients. Additionally, a CMR finding of morphological (e.g., enlarged ventricles) or functional post-viral cardiac injury represents an absolute indication for HF treatment.

## 5. Conclusions

Patients with PASC-CVS have a high incidence of CMR abnormalities. Improvement in cardiovascular symptoms and CMR findings might be attributed to NSAID maintenance and HF therapy. However, a randomized placebo-controlled study should be performed to confirm our findings.

## Figures and Tables

**Figure 1 biomedicines-11-03312-f001:**
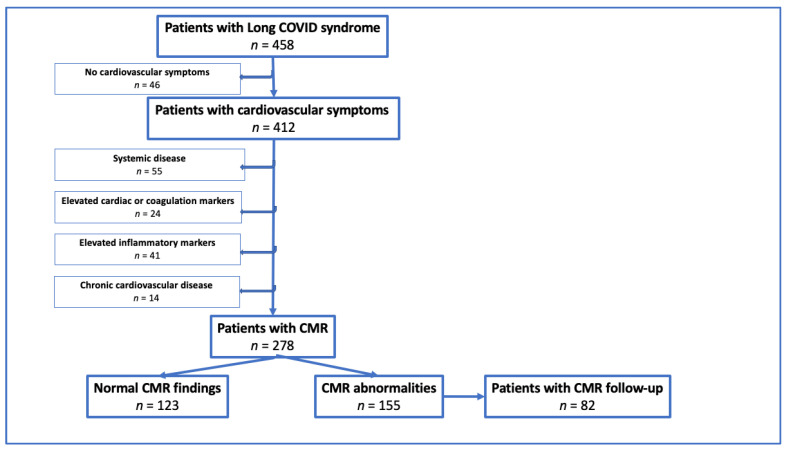
Flow chart of the study. CMR: cardiac magnetic resonance imaging.

**Figure 2 biomedicines-11-03312-f002:**
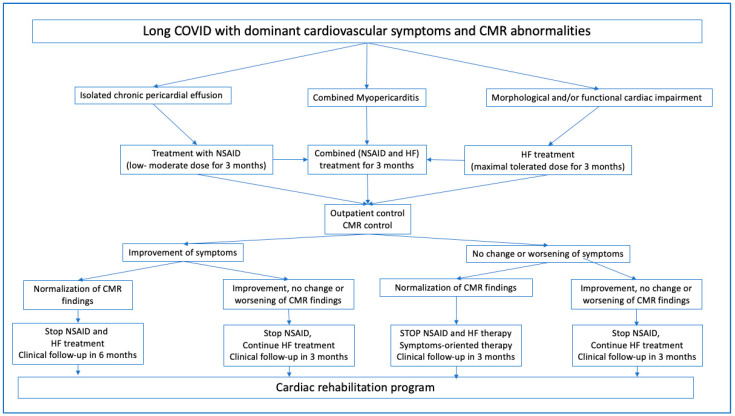
Therapy regimen for patients with long COVID-19 syndrome and cardiovascular symptoms and cardiac magnetic resonance imaging (CMR) abnormalities. NSAID: non-steroidal anti-inflammatory drug; HF: heart failure.

**Figure 3 biomedicines-11-03312-f003:**
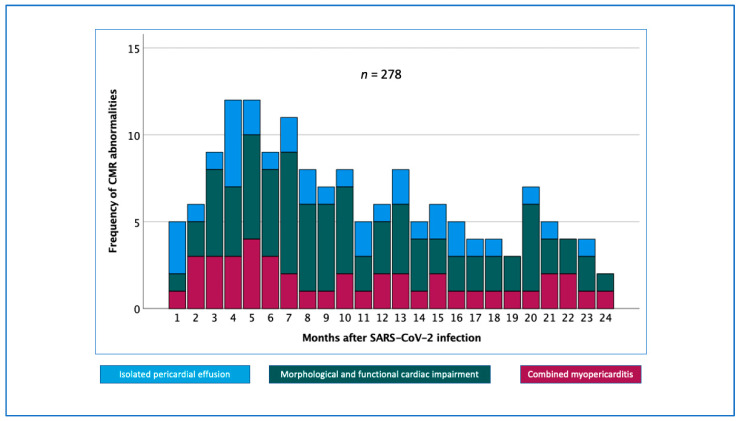
Time-dependent frequencies of cardiac magnetic resonance imaging (CMR) abnormalities.

**Figure 4 biomedicines-11-03312-f004:**
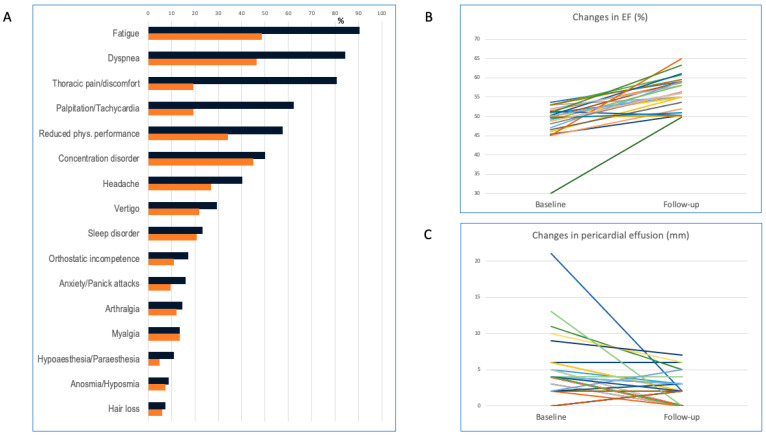
Improvement in clinical symptoms, increase in left ventricular ejection fraction, and decrease in pericardiac effusion after medical treatment in patients with long COVID-19 syndrome. (**A**) Frequency of symptoms at first (black columns) clinical presentation and after 3 months of medical treatment (orange columns); (**B**) individual changes in left ventricular ejection fraction in patients with reduced left ventricular function at first clinical presentation (*n* = 25); and (**C**) pericardial effusion (*n* = 51) between baseline and follow-up.

**Figure 5 biomedicines-11-03312-f005:**
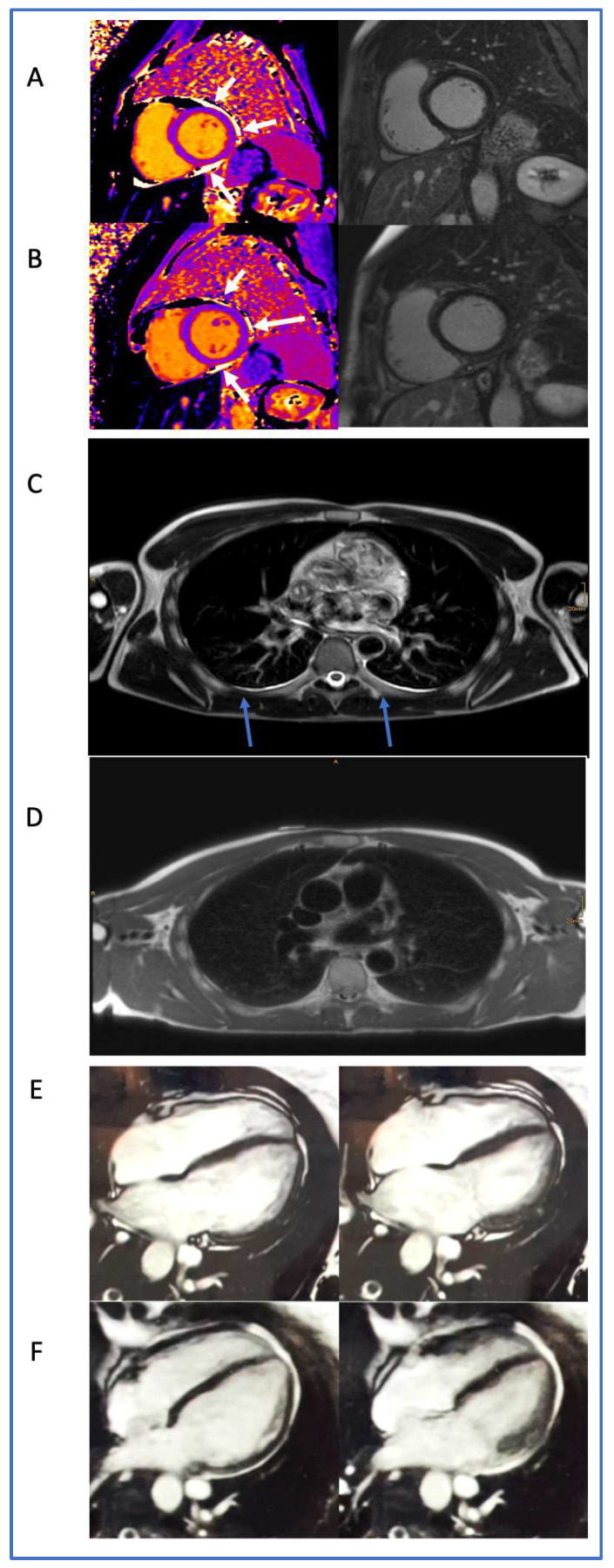
Cardiac magnetic resonance imaging (CMR) of patients with cardiovascular long COVID-19 syndrome at first clinical presentation and after treatment at control. (**A**) Non-physiological pericardial effusion (white arrows) and (**B**) regression over a 3 month treatment period in a 53-year-old male patient. (**C**) Pleural effusion (blue arrows) at first clinical presentation in a 32-year-old male patient with long COVID-19 syndrome, with (**D**) complete regression after treatment 3 months later (bottom). (**E**) Baseline end-diastolic (left) and end-systolic (right) images of the left ventricle in a 42-year-old woman, with a left ventricular ejection fraction (LVEF) of 45%, and (**F**) 3 months later, after treatment with an LVEF of 52% calculated by CMR.

**Table 1 biomedicines-11-03312-t001:** Clinical, electrocardiographic, and laboratory data in the long COVID-19 cohort with cardiac symptoms after COVID-19 infection.

Clinical Characteristics and Laboratory Findings	*n* = 278
Time between COVID-19 infection and CMR (days)	328 ± 214
Number of patients with at least one COVID-19 vaccine (prior to or after COVID-19 infection)	227 (81.7%)
Anti-spike protein titer (AU/mL)	1546 ± 1093
Female sex *n* (%)	196 (70.5%)
Age (years)	43 ± 13
Body mass index (kg/m^2^)	25.2 ± 5.2
Diabetes mellitus *n* (%)	7 (2.5%)
Hypertension *n* (%)	44 (15.8%)
Hyperlipidemia *n* (%)	39 (14.0%)
Smoking *n* (%)	9 (3.2%)
Systolic blood pressure (mmHg)	129 ± 17
Diastolic blood pressure (mmHg)	82 ± 11
Heart rate (bpm)	73 ± 12
Cumulative ECG abnormalities *n* (%)	66 (23.7%)
Cardiac arrhythmias *n* (%)	13 (4.7%)
Conduction abnormalities *n* (%)	59 (21.2%)
QRS width (ms)	89.2 ± 12.7
Leukocyte (G/L)	7.0 ± 2.1
Fibrinogen (mg/dL)	313 ± 67
D-dimer (ug/mL)	0 (0;0.37)
Cardiolipin IgG (U/mL)	1.2 (0;1.7)
Cardiolipin IgM (U/mL)	1.4 (0;2.3)
Creatine kinase (U/L)	89 (62;122)
Creatine kinase myocardial subfraction (U/L)	14.1 (12;18.5)
Troponin T (ng/L)	0 (0;6)
NT-proBNP (pg/mL)	44.0 (23.7;82.0)
C-reactive protein (mg/dL)	0.08 (0.04;0.20)
Rheumafactor latex (IU/mL)	0.0 (0.0;0.0)
Alpha 1 antitrypsin (mg/dL)	138 ± 24.2
Interleukin-6 (pg/mL)	0 (0;2.14)
Procalcitonin (ng/mL)	0.03 (0;0.04)
Transferrin (mg/dL)	267.5 ± 45.6
Transferrin saturation (%)	24.3 ± 10.4

Values are given as mean ± SD, median with interquartile range, or *n* (%).

**Table 2 biomedicines-11-03312-t002:** Cardiac magnetic resonance (CMR) findings in the long COVID-19 cohort with cardiac symptoms after COVID-19 infection.

CMR Findings	*n* = 278
No abnormalities	123 (44.2%)
Cumulative CMR abnormalities *n* (%)	155 (55.8%)
Isolated pericardial effusion (without functional impairment) *n* (%)	34/278 (12.2%)
Morphological and functional impairment *n* (%)	79/278 (28.4%)
Combined myopericarditis *n* (%)	42/278 (15.1%)
Pericardial effusion (w/wo functional impairment) *n* (%)	72 (25.9%)
Reduced LVF *n* (%)	39 (14.0%)
Reduced RVF *n* (%)	55 (19.8%)
Biventricular enlargement *n* (%)	21 (7.6%)
Isolated LV enlargement *n* (%)	56 (20.1%)
Isolated RV enlargement *n* (%)	47 (16.9%)
Myocardial edema *n* (%)	9 (3.2%)
T1 increase *n* (%)	5 (1.8%)
Nonischemic late gadolinium enhancement	35 (12.6%)
Pleural effusion *n* (%)	16 (5.8%)
CMR LV EF (%)	59 ± 7
CMR LV EDV (mL)	142 ± 36
CMR LV ESV (mL)	60 ± 20
CMR RV EF (%)	55 ± 6
CMR RV EDV (mL)	152 ± 39
CMR RV ESV (mL)	70 ± 25

Abbreviations: LV, left ventricular; RV, right ventricular; LVF, left ventricular function; RVF, right ventricular function; EF, ejection fraction; EDV, end-diastolic volume; ESV, end-systolic volume.

**Table 3 biomedicines-11-03312-t003:** Baseline and follow-up cardiac magnetic resonance (CMR) findings in patients with dominant cardiovascular syndromes and CMR abnormalities at baseline and after medical treatment.

CMR Findings Pre- and Post-Treatment	Baseline (*n* = 82)	Follow-up (*n* = 82)	*p* Value
Cumulative CMR abnormalities *n* (%)	82 (100%)	53 (64.6%)	<0.001
CMR phenotype *n* (%)			<0.001
No abnormalities *n* (%)		29 (35.4%)	
Isolated pericardial effusion (without functional impairment) *n* (%)	19 (23.2%)	22 (26.8%)	
Morphological and functional impairment *n* (%)	29 (35.4%)	20 (24.4%)	
Combined myopericarditis *n* (%)	34 (41.5%)	11 (13.4%)	
Pericardial effusion (w/wo functional impairment) *n* (%)	51 (62.2%)	32 (39.0%)	0.005
Reduced LVF *n* (%)	25 (30.5%)	4 (4.9%)	<0.001
Reduced RVF *n* (%)	27 (32.9%)	11 (13.4%)	0.039
Biventricular enlargement *n* (%)	14 (17.1%)	9 (11.0%)	
Isolated LV enlargement *n* (%)	29 (35.4%)	16 (19.5%)	0.035
Isolated RV enlargement *n* (%)	18 (22.0%)	16 (19.5%)	
Myocardial edema *n* (%)	8 (9.8%)	1 (1.2%)	0.034
T1 increase *n* (%)	1 (1.2%)	0 (0%)	
Nonischemic late gadolinium enhancement *n* (%)	20 (24.4%)	16 (19.5%)	
Pleural effusion *n* (%)	5 (6.2%)	8 (9.8%)	
Pericardial effusion (mm) *	4 (3;5.75)	2 (0;3)	<0.001
CMR LV EF (%)	57 ± 7	59 ± 5	0.034
CMR LV EDV (mL)	150 ± 39	151 ± 36	
CMR LV ESV (mL)	65 ± 21	64 ± 20	
CMR RV EF (%)	53 ± 7	557 ± 7	
CMR RV EDV (mL)	161 ± 44	167 ± 44	
CMR RV ESV (mL)	76 ± 25	76 ± 29	

* Median with interquartile range and Wilcoxon test. Abbreviations: LV, left ventricular; RV, right ventricular; LVF, left ventricular function; RVF, right ventricular function; EF, ejection fraction; EDV, end-diastolic volume; ESV, end-systolic volume.

**Table 4 biomedicines-11-03312-t004:** Changes in CMR abnormalities 3 months after the recommended therapy.

Subgroups of CMR Phenotypes/Follow-up CMR Findings after Treatment	Normalized	Improved	Unchanged	Worsened	Total
Isolated pericardial effusion (without functional impairment) *n* (%)	6 (31.6%)	5 (26.3%)	5 (26.3%)	3 (15.8%)	19 (100%)
Morphological and functional impairment *n* (%)	12 (41.4%)	7 (24.1%)	8 (27.6%)	2 (6.9%)	29 (100%)
Combined myopericarditis *n* (%)	11 (32.4%)	22 (64.7%)	1 (2.9%)	0 (0%)	34 (100%)
Total	29 (35.4%)	34 (41.5%)	14 (17.1%)	5 (6.1%)	82 (100%)

## Data Availability

The data underlying this article cannot be shared publicly due to the small cohort and the possible risk of identification of individual patients. The data will be shared on reasonable request to the corresponding author.

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
