# Peer review of "Improvement of Symptoms and Cardiac Magnetic Resonance Abnormalities in Patients with Post-Acute Sequelae of SARS-CoV-2 Cardiovascular Syndrome (PASC-CVS) after Guideline-Oriented Therapy"

_biomedicines, 2023, doi:10.3390/biomedicines11123312_

Round 1

Reviewer 1 Report

Comments and Suggestions for Authors

Long COVID and  post-covid become an important issue of daily clinical practice after pandemic.

Team from Vienna University presented an elegant study  of  patients who experienced some kind of discomfort almost a year after COVID-19 disease. 55.8% of  those  patients were found to have minor abnormalities on CMR that disappeared in 3-4  moths CMR follow-up in most patients. Probably due to treatment or due to natural course.

The are several major concerns.

1.       Where is no placebo arm or general treatment arm. Thus we can’t conclude that it is the result of the treatment . Have to be mentioned in the limitation section. Conclusion could be corrected

2.       Figure 2 is the key to the results. 4 groups after follow- up could be analyzed: symptoms resolutions  with CMR resolution, symptoms + CMR-, symptoms- CMR+, symptoms – CMR-   

Minor issue:

1.       Figure 2 have  to incorporate numbers in every box for clarification

2.       What does myocarditis in Table 2 means.  According  to Lake Louise criteria ?

3.       No data on LGE in table 2.

4.       Figure 4 is difficult to understand. A and B could be presented on the same picture.

Comments on the Quality of English Language

Editing is required

Author Response

We would like to thank the reviewer for their valuable comments, which have led to us improving the paper.

Reviewer 1

Long COVID and  post-covid become an important issue of daily clinical practice after pandemic.Team from Vienna University presented an elegant study  of  patients who experienced some kind of discomfort almost a year after COVID-19 disease. 55.8% of  those  patients were found to have minor abnormalities on CMR that disappeared in 3-4  moths CMR follow-up in most patients. Probably due to treatment or due to natural course.The are several major concerns.

  1. Where is no placebo arm or general treatment arm. Thus we can’t conclude that it is the result of the treatment . Have to be mentioned in the limitation section. Conclusion could be corrected

Many thanks for the Reviewer to point out this important issue. Accordingly, we have added the statement into the Limitation section as follows:

Due to lack of control or placebo group, the efficacy of the suggested therapy might be overestimated. However, many patients had persistent CMR abnormalities even longer than 1 year, which has been resolved or improved after the therapy, parallel with the decrease in cardiovascular symptoms in our cohort, which suggest the beneficial effect of our therapy. Considering the psychical vulnerability of the patients with PASC-CVS, a blinded study with eventual randomization to placebo arm was not accepted by our patients. Additionally, a CMR finding on morphological (eg. enlarged ventricles) or functional post-viral cardiac injury represent an absolute indication for HF treatment.

We have also corrected the Conclusion (abstract) as follows:

Clinical symptoms improved markedly with a decrease in CMR abnormalities, which might be attributed to the maintenance of NSAID and HF medical treatment of PASC-CVS.

and Conclusions (text):

Patients with PASC-CVS have high incidence of CMR abnormalities. Improvement in cardiovascular symptoms and CMR findings might be attributed to the NSAID maintenance and HF therapy. However, randomized placebo-controlled study should be performed to confirm our findings.

In addition, we agree with the Reviewer, that post-viral non-significant pericardial effusion does not necessarily require medical therapy, but, however, as mentioned in the Introduction:

„the 6-month mortality was significantly higher in SARS-CoV-2-infected patients if they had pericarditis, compared with COVID-19 positive patients without pericarditis, even with a small amount of pericardial effusion [9, 10].“

  1. Figure 2 is the key to the results. 4 groups after follow- up could be analyzed: symptoms resolutions  with CMR resolution, symptoms + CMR-, symptoms- CMR+, symptoms – CMR-   

Thank you for the comment. Accordingly, we have corrected the Figure 2, and added an additional Table (Table 4) with the incidences of normalization, improvement, no change or worsening of the CMR findings of the different CMR phenotype groups.

Minor issue:

  1. Figure 2 have  to incorporate numbers in every box for clarification

Please see corrected Figure 2 and the newly added Table 4 and our comment above. Adding the numbers into the Figure 2 would unnecessarily complicate the Figure. As mentioned, we have added the numbers in the Table 4.

  1. What does myocarditis in Table 2 means.  According  to Lake Louise criteria ? 

Thank you for the comment. We have corrected the term, from myocarditis to myocardial edema. Since our patients had no signs or clinical or laboratory suspect of acute myocarditis, no CMR abnormalities fulfilled the Lake Louse and modified Lake Louise criteria. We have added the following comment into the Discussion:

At the clinical presentations, acute viral myocarditis was excluded in all patients, based on normal cardiac enzymes and normal inflammatory parameter and lack of ECG signs or clinical symptoms of acute myocarditis. Few patients presented isolated myocardial edema, or T1 increase or non-ischemic late gadolinium enhancement in the CMR imaging, indicating chronic myocardial injury. However, by lack of supportive acute clinical scenario and laboratory signs, these CMR findings did not fulfill the modified Lake Louse criteria of acute myocarditis (Refs 29,30). In addition, no CMR was performed during the acute phase of the SARS-CoV-2 infection.

In accordance with the myocarditis criteria, we have corrected the CMR phenotypes in the chronic phase of SARS-CoV-2 infection, adding the morphological changes (eg. enlarged ventricles, or T1-increase or late gadolinium enhancement) to the functional impairment, leading to reclassification of few patients, with corresponding re-evaluating some CMR data, which, however did not influence any results and conclusions.

  1. No data on LGE in table 2. 

In accordance with the suggestion of the Reviewer, we have added the LGE to the Table 2.

  1. Figure 4 is difficult to understand. A and B could be presented on the same picture.

Thank you for the comment. We have modified the Figure 4 as the Reviewer suggested.

Editing is required

The paper has undergone language editing.

Reviewer 2 Report

Comments and Suggestions for Authors

Biomedicines 2742243: Review

Improvement of symptoms and cardiac magnetic resonance abnormalities in PASC-CVS patients after guideline-oriented therapy

By Mariann GyÖngyÖsi, Ena Hasimbegovic, Emilie Han, Katrin Zlabinger, Andreas Spannbauer, Martin Riesenhuber, Kevin Hamzaraj, Jutta Bergler-Klein, Christian Hengstenberg, Andreas Kammerlander, Stefan Kastl, Christian Loewe, Dietrich Beitzke

It has been shown that cardiac magnetic resonance (CMR) studies demonstrated abnormalities in patients with mild-moderate SARS-CoV-2 infection, suggesting on-going myocardial inflammation. In an effort to characterize chronic abnormalities in these patients evaluated by CMR patients with post-acute sequelae of SARS-CoV-2 cardiovascular syndrome (PASC-CVS) who were included prospectively into the Vienna POSTCOV Registry between March 2021 and March 2023 (n=278, 43±13 years, 70.5% female). Clinical, laboratory and CMR findings were recorded, and those with abnormal results were classified into isolated chronic pericardial (±pleural) effusion, isolated cardiac function impairment or both (myopericarditis) groups. Medical treatment included non-steroidal anti-inflammatory agent (NSAID) for pericardial effusion and condition-adapted maximal dose of heart failure (HF) treatment. Three months after medical therapy, clinical assessment and CMR was repeated in 82 patients whose laboratory analyses were normal. CMR abnormalities at baseline were found in 155 patients (55.8%). Condition-adapted HF treatment led to a significant increase in left ventricular ejection fraction (LVEF) in patients with initially reduced LVEF (from 49±5% to 56±4%, P=0.009, n=25). Low-moderate doses of NSAIDs for 3 months significantly reduced pericardial effusion (P<0.001, n=51). Clinical symptoms improved markedly with a decrease in CMR abnormalities.

Maintenance NSAID medical treatment of PASC-CVS led to significant improvement in cardiac function and decreased pericardial effusion, and were associated with improvements in PASC-CVS.

Major comments

The findings are interesting but there seem to be a contention that the therapy administered was the cause of improvement as documented at the 3 month follow up. There is an emphasis on the therapeutic protocol. I can accept more that heart failure therapy is indicated in the face of LV systolic dysfunction and that it is probable that this therapy is helpful and effective with regard to improvement. Less so in the case of pericardial effusion; though it cannot be ruled out that NSAID with or without steroids aided to eliminate the fluid. We know that oftentimes pericardial fluid following a benign viral infection is absorbed spontaneously some week later even without specific therapy. In summary, the authors outline a therapeutic protocol and prematurely ascribe the improvement to this approach. See line 28-29:

“Maintenance NSAID medical treatment of PASC-CVS led to significant improvement in cardiac function and decreased pericardial effusion, and were associated with improvements in PASC-CVS”. Such a statement is not valid in the absence of a control group!  

I have to add that within the discussion section the authors do state that “the role of spontaneous improvement of cardiac abnormalities or the role of arbitrary intake of diverse anti-inflammatory or antioxidant dietary supplements during the long COVID period cannot be excluded.”

Minor comments

(1)  Line 24-25: “from 48.8±4.7 % to 56.2±4.2%, P=0.009, n=25)”. It is highly pretentious to present LVEF with decimal values when taking into account the method of determining this value. Should be ” from 49±5 % to 56±4%, P=0.009…..”

(2)  Line 123: What is “fluidum”?

Comments on the Quality of English Language

Acceptable. Line 123: What is “fluidum”?

Author Response

We would like to thank the reviewer for their valuable comments, which have led to us improving the paper.

Reviewer 2.

Major comments

The findings are interesting but there seem to be a contention that the therapy administered was the cause of improvement as documented at the 3 month follow up. There is an emphasis on the therapeutic protocol. I can accept more that heart failure therapy is indicated in the face of LV systolic dysfunction and that it is probable that this therapy is helpful and effective with regard to improvement. Less so in the case of pericardial effusion; though it cannot be ruled out that NSAID with or without steroids aided to eliminate the fluid. We know that oftentimes pericardial fluid following a benign viral infection is absorbed spontaneously some week later even without specific therapy. In summary, the authors outline a therapeutic protocol and prematurely ascribe the improvement to this approach. See line 28-29:

“Maintenance NSAID medical treatment of PASC-CVS led to significant improvement in cardiac function and decreased pericardial effusion, and were associated with improvements in PASC-CVS”. Such a statement is not valid in the absence of a control group!  

I have to add that within the discussion section the authors do state that “the role of spontaneous improvement of cardiac abnormalities or the role of arbitrary intake of diverse anti-inflammatory or antioxidant dietary supplements during the long COVID period cannot be excluded.”

Many thanks for the important comment of the Reviewer. We agree, that the lack of control or placebo group would be necessary for objective assessment of the therapy efficacy. However, as mentioned above, our patients had persistent CMR abnormalities with persistent symptoms even longer than 1 year after the viral infection. The resolution or improvement of the symptoms parallel with decrease in the pathologies in the CMR images supports or hypothesis. However, as Reviewer 1 also requested, we have added some additional sentences into the Limitation section, and corrected the Conclusion in the Abstract and in the main text.

Minor comments

  • Line 24-25: “from 48.8±4.7 % to 56.2±4.2%, P=0.009, n=25)”. It is highly pretentious to present LVEF with decimal values when taking into account the method of determining this value. Should be ” from 49±5 % to 56±4%, P=0.009…..” 

In accordance with the Reviewer`s suggestion, we have corrected the numbers In Tables and text.

  • Line 123: What is “fluidum”?

We have corrected the fluidum to fluid. We have added some comments regarding combined pericardial and pleural fluids, as follows:

Polyserositis is characterized by inflammation and effusion of the serous membranes, e.g. pleaura, pericardium and peritoneum. Combined pericarditis and pleuritis are the most common appearance of polyserositis, albeit the diagnosis is difficult with lack of diagnostic and therapeutic guidelines (Refs 31,32). Some case reports emphasize the diagnostic difficulties of polyserositis in patients,  especially children with multiorgan syndromes with acute SARS-CoV-2 infection (Ref 33). We have detected simultaneous pericardial and pleural effusion in 16 patients. The long-time maintenance of the polyserositis after the acute infection suggest either a chronic inflammation or autoimmune reaction; both processes require clinical controls.

Round 2

Reviewer 1 Report

Comments and Suggestions for Authors

The  manuscriopt is improved and can be reccommended for the  Journal

Comments on the Quality of English Language

English is  fine

Author Response

We would like to thank the Reviewer for investing time and effort to review our paper.

Reviewer 2 Report

Comments and Suggestions for Authors

In general I suggest further English editing to improve the style of the paper, but I recommend acceptance of the paper. I simply do not have the time for meticulous editing of the paper.  

With regard to the title of the paper I suggest to use the explicit term of

PASC-CVS, i.e., Post-acute sequelae of SARS-CoV-2 cardiovascular syndrome

and not PASC-CVS, which most readers are unfamiliar with.

Page 2, line 54: “With a lack of signs….” better “in the absence…”

Line 293: “Lake Louse…”, or Lake Louise…”?

Line 363: “psychical vulnerability…”, or better “psychologically vulnerable…”.

Comments on the Quality of English Language

I think the paper is to be accepted after some English editing, it is not bad but still needs polishing. 

Author Response

We would like to thank the reviewer for the valuable comments, which have led to us improving the paper.

In general I suggest further English editing to improve the style of the paper, but I recommend acceptance of the paper. I simply do not have the time for meticulous editing of the paper. 

Many thanks for this comment of the Reviewer. Before first resubmission, the paper underwent a language editing via the MDPI English Editing Service (file:///Users/mariann/Downloads/English-Editing-Certificate-74741-1.pdf).

However, as the Reviewer has noticed, there is still a typo regarding the Lake Louise criteria; we have corrected the term in the revised paper.

In accordance with the Reviewer suggestion, we have also changed the following terms:

Page 2, line 54: “ “in the absence…”

Line 293: “ Lake Louise…”

Line 363:  “psychological vulnerability …”.

With regard to the title of the paper I suggest to use the explicit term of PASC-CVS, i.e., Post-acute sequelae of SARS-CoV-2 cardiovascular syndrome and not PASC-CVS, which most readers are unfamiliar with.

In accordance with the Reviewer suggestion, we have modified the title as follows:

“Improvement of symptoms and cardiac magnetic resonance abnormalities in patients with Post-acute sequelae of SARS-CoV-2 cardiovascular syndrome (PASC-CVS) after guideline-oriented therapy”

We have used the PASC-CVS term in the manuscript in accordance with the 2022 ACC Expert Consensus Decision Pathway on Cardiovascular Sequelae of COVID-19 in Adults (Ref 3)